# Epidermal Growth Factor Receptor-Targeted Neoantigen Peptide Vaccination for the Treatment of Non-Small Cell Lung Cancer and Glioblastoma

**DOI:** 10.3390/vaccines11091460

**Published:** 2023-09-05

**Authors:** Fenge Li, Huancheng Wu, Xueming Du, Yimo Sun, Barbara Nassif Rausseo, Amjad Talukder, Arjun Katailiha, Lama Elzohary, Yupeng Wang, Zhiyu Wang, Gregory Lizée

**Affiliations:** 1Core Laboratory, Tianjin Beichen Hospital, Tianjin 300400, China; rosetea85@163.com; 2Department of Oncology, Tianjin Beichen Hospital, Tianjin 300400, China; 3Department of Neurosurgery, Tianjin Beichen Hospital, Tianjin 300400, China; 4Department of Melanoma Medical Oncology, The University of Texas MD Anderson Cancer Center, Houston, TX 77054, USA; 5Department of Immuno-Oncology, The Fourth Hospital of Hebei Medical University, Shijiazhuang 050011, China; 6Department of Immunology, The University of Texas MD Anderson Cancer Center, Houston, TX 77054, USA

**Keywords:** epidermal growth factor receptor, peptide vaccine, lung cancer, immunotherapy

## Abstract

The epidermal growth factor receptor (EGFR) plays crucial roles in several important biological functions such as embryogenesis, epithelial tissue development, and cellular regeneration. However, in multiple solid tumor types overexpression and/or activating mutations of the *EGFR* gene frequently occur, thus hijacking the EGFR signaling pathway to promote tumorigenesis. Non-small cell lung cancer (NSCLC) tumors in particular often contain prevalent and shared EGFR mutations that provide an ideal source for public neoantigens (NeoAg). Studies in both humans and animal models have confirmed the immunogenicity of some of these NeoAg peptides, suggesting that they may constitute viable targets for cancer immunotherapies. Peptide vaccines targeting mutated EGFR have been tested in multiple clinical trials, demonstrating an excellent safety profile and encouraging clinical efficacy. For example, the CDX-110 (rindopepimut) NeoAg peptide vaccine derived from the EGFRvIII deletion mutant in combination with temozolomide and radiotherapy has shown efficacy in treating EGFRvIII-harboring glioblastoma multiforme (GBM) patients undergone surgery in multiple Phase I and II clinical trials. Furthermore, pilot clinical trials that have administered personalized NeoAg peptides for treating advanced-stage NSCLC patients have shown this approach to be a feasible and safe method to increase antitumor immune responses. Amongst the vaccine peptides administered, EGFR mutation-targeting NeoAgs induced the strongest T cell-mediated immune responses in patients and were also associated with objective clinical responses, implying a promising future for NeoAg peptide vaccines for treating NSCLC patients with selected EGFR mutations. The efficacy of NeoAg-targeting peptide vaccines may be further improved by combining with other modalities such as tyrosine kinase or immune checkpoint inhibitor (ICI) therapy, which are currently being tested in animal models and clinical trials. Herein, we review the most current basic and clinical research progress on EGFR-targeted peptide vaccination for the treatment of NSCLC and other solid tumor types.

## 1. Introduction

Growth factors are hormone-like molecules that can promote cell division and migration but are also involved in tumor growth and metastasis. Dr. Rita Levi-Montalcini discovered the first growth factor, nerve growth factor (NGF), in 1954 [1], and epidermal growth factor (EGF) was discovered by Dr. Stanley Cohen in 1965 [2]. For their pioneering work, these two scientists shared the 1986 Nobel Prize in Physiology/Medicine. In 1982, Dr. Stanley Cohen and colleagues reported the identification of the cell surface receptor of EGF [3], called the human epidermal growth factor receptor (EGFR), also known as Erythroblastic oncogene B1 (ErbB1).

Epidermal growth factor receptor (EGFR) is a ~170 kDa protein encoded by the *EGFR* gene located on chromosome 7p12.2. EGFR belongs to the ErbB family comprised of EGFR/ErbB-1, HER2/ErbB-2, HER3/ErbB-3 and HER4/ErbB-4. All members of the ErbB family are transmembrane receptor tyrosine kinases (RTKs) which consist of an extracellular ligand-binding domain, a transmembrane domain, an intracellular tyrosine kinase (TK) domain, and a regulatory tail region containing conserved phosphorylation sites (Figure 1) [4]. ErbB family RTKs bind to extracellular growth factor ligands, which in turn initiate downstream signaling pathways to exert critical functions involving cell proliferation, survival, and differentiation [5,6].

## 2. Basic Molecular Function of EGFR

When the extracellular ligand-binding domain of EGFR is not bound to its ligand, this protein exists as a monomer on the plasma membrane (PM) and the tyrosine residues located in the intracellular TK domain and regulatory tail domain are not phosphorylated [4,7,8]. In this inactive state, downstream signaling is not triggered. However, upon binding to its ligand, EGFR undergoes a series of conformational changes, leading to its homodimerization or heterodimerization with other ErbB members [7,9]. This allows the formation of an asymmetric dimer between the two juxtaposed intracellular TK domains in which one kinase domain (called the “activator kinase”) allosterically activates the other (“receiver kinase”) by trans-phosphorylating its critical intracellular tyrosine residues [4,10]. Thereafter, the receiver kinase also activates the activator kinase through an trans-autophosphorylation mechanism. The newly phosphorylated tyrosine residues serve as attachment sites for multiple adaptors (e.g., GRB2), cytoplasmic enzymes (e.g., PLC-gamma) or transcription factors (e.g., STAT3), thereby triggering downstream signaling cascades [5,8] (Figure 1). The major signaling pathways triggered by EGFR activation include the Ras/MAPK pathway, the PI3K/AKT pathway, and the JAK/STAT pathway [7,8]. Those signaling pathways are largely pro-survival and anti-apoptotic, playing crucial roles in cell survival, proliferation, differentiation, and motility [4,11].

Similar to many other receptors, EGFR is internalized through multiple endocytic pathways with different features that regulate the activity and fate of this receptor [8,12]. This includes clathrin-mediated endocytosis (CME) and several nonclathrin endocytic (NCE) pathways involved in EGFR internalization [8,13]. Because EGFR-mediated signaling governs cell survival, proliferation and other key events, the compartmentalization and trafficking of EGFR are strictly regulated in human cells to prevent neoplastic transformation. Firstly, the structural rearrangement and phosphorylation of EGFR TK domains induced by ligand binding also promotes the recruitment of endocytic machinery that mediates EGFR endocytosis to downregulate signaling, with internalization rates ~10-fold higher than that of inactive EGFR [14]. Secondly, once activated several critical lysine residues within the EGFR TK domain undergo ubiquitination by the E3 ligase Cbl in complex with the adaptor molecule Grb2 [15,16], which serves as a signal for receptor internalization into the NCE pathway, and subsequently for targeting EGFR for lysosomal degradation [16].

EGFR has seven different known ligands, including EGF, transforming growth factor-alpha (TGF-α), heparin-binding EGF-like growth factor (HBEGF), betacellulin (BTC), amphiregulin (AREG), epiregulin (EREG), and epigene (EPGN) [17]. Binding different ligands leads to diverse cellular responses and intracellular trafficking events [18], which may be attributed to their ability to differentially stabilize the EGFR dimers [13], therefore determining specific signaling outputs. Among these ligands, EGF, TGF-α, HBEGF and BTC are high-affinity ligands, whereas AREG, EREG, and EPGN constitute low-affinity ligands [4,8]. The different strength of the ligand–receptor interaction dictates whether the ligand dissociates from the receptor in the mildly acidic environment of the endosomes [13].

Although the best characterized functions of the EGFR are aforementioned ligand- and kinase-dependent activation, also called the “canonical” EGFR signaling pathway [8], novel functions have been identified recently. These “noncanonical” EGFR signaling pathways induced by cellular and environmental stresses can be independent of its TK activity, playing indispensable roles in the regulation of autophagy and metabolism [8,19]. EGFR signaling has been involved in regulating several metabolic processes critical for cell proliferation, including fatty acid and pyrimidine biosynthesis, and glucose catabolism [20].

## 3. The Role of EGFR in Cancer Development

Due to the central role of EGFR-mediated signaling in promoting cell survival, proliferation and other key functions, once deregulated it can contribute to the activation of critical oncogenic pathways, ultimately leading to uncontrolled cell proliferation, tumor invasiveness, metastasis, angiogenesis and other malignant phenotypes [4,21,22]. EGFR signaling is frequently hijacked by cancer cells to promote oncogenesis, mainly through *EGFR* gene amplification and/or overexpression, in addition to a variety of activating mutations in different types of human cancers [4,11]. Approximately 60% of East Asian non-small-cell lung cancers (NSCLC), 30% of breast cancers, and 40% of glioblastoma multiforme (GBM) either overexpress or bear activating mutations/in-frame deletions in EGFR or its family members [8,22,23,24].

*EGFR* gene amplification or overexpression causes an increase in EGFR density at the PM, which enhances the formation and accumulation of receptor homo- and heterodimers, leading to excessive EGFR kinase activity and constitutively active signaling [25,26,27]. Increased EGFR levels also cause saturation of the endocytic and/or the ubiquitination machinery, which reinforces the sustained signaling in cancer cells [28]. In addition, EGFR/HER2 heterodimers can evade negative regulation mechanisms including ubiquitination and degradation [29], resulting in most of these receptor dimers being recycled back to the PM, leading to abnormal accumulation of EGFR and overactive signaling.

Mutations occurring in the *EGFR* gene are typically gain-of-function and drive tumorigenesis. They usually mimic the ligand-activated wild-type (WT) form, but their tyrosine phosphorylation status is significantly lower than that of ligand-activated WT EGFR [30]. This relatively low but much more persistent EGFR activity results in sustained and enhanced signaling, leading to oncogenesis. The most frequent cancer-associated mutations in the *EGFR* gene are illustrated in Figure 2 and will be discussed in detail in the next section. Oncogenic mutations located in the extracellular domain of EGFR cause ligand-independent receptor activation and sustained downstream signaling. For example, the EGFRvIII (aka EGFRΔvIII) mutant, present in 25–33% of all patients with GBM and representing the most frequent genetic aberration in brain tumors, contains a deletion of exons 2–7 of the *EGFR* gene, leading to the loss of large parts of the extracellular ligand-binding domain, and hence cannot bind any known ligands [24,31]. However, EGFRvIII displays low-level ligand-independent constitutive signaling that is augmented by reduced internalization and degradation [25,32]. In vitro and in vivo evidence has supported the direct driving force of EGFRvIII in neoplastic transformation [24]. Oncogenic mutations located in the intracellular TK domain (e.g., L858R and exon 19 E746_A750 deletion) disrupt the recruitment site of the Cbl E3 ligase, which attenuates EGFR ubiquitination and lysosomal degradation, thereby contributing to increased signaling properties. The mechanism underlying the impairment of Cbl recruitment caused by these mutations may include the hypophosphorylation of the direct Cbl-binding site Y1045 residue in the intracellular domain. However, L858R, one of the most frequent mutations in lung cancers, actually enhances the phosphorylation of direct Cbl-binding sites. It has been proposed that this mutation downregulates Cbl recruitment and EGFR degradation by enhancing the formation of EGFR/HER2 heterodimers which can evade the Cbl-mediated EGFR ubiquitination and degradation [28,30]. Because cancer cells with these genetic alterations in *EGFR* gene become highly dependent on the continuous activation of the EGFR pathway to establish and maintain their growth advantage, EGFR has been a key target of multiple cancer therapies in clinical practice. Objective responses of lung cancer to EGFR tyrosine kinase inhibitors (TKIs) are significantly associated with mutations in *EGFR* exons 18–21 [11,22]. The most frequent of these mutations are in-frame deletions in exon 19 and the L858R point mutation in exon 21, accounting for 85~90% among all TK domain mutations [23].

Inappropriate activation of EGFR in cancer can also originate from the changes occurring in other genes involved in receptor endocytosis and recycling. Overexpression and amplification of genes encoding endocytic/recycling molecules that regulate EGFR internalization, for example, some Rab family GTPases and 5′-inositol lipid phosphatase synaptojanin 2 (SYNJ2), alter EGFR trafficking and degradation that causes sustained EGFR activation, leading to oncogenesis [33].

In tumors bearing EGFR amplification and/or activating mutations, cancer cells become addicted to the oncogenic function of EGFR, making EGFR an attractive therapeutic target. A number of small-molecule inhibitors and antibodies have been developed to specifically target this protein. Multiple EGFR tyrosine kinases inhibitors (TKIs) such as erlotinib, gefitinib, afatinib, and osimertinib have been used in first-line treatment for advanced NSCLC patients with activating EGFR mutations [34]. These agents reversibly bind the ATP-binding site of the TK domain of EGFR, with higher binding affinity for TK domains bearing activating mutations compared to the wild-type version, thereby inhibiting its kinase activity. EGFR TKIs have shown significant improvement in both efficacy and safety in the treatment of patients with EGFR-mutant advanced NSCLC compared to standard platinum-based chemotherapy. Despite an initial response to EGFR TKIs, almost all patients develop tumor progression and acquired resistance to EGFR TKIs within one to two years [34]. The mechanisms of drug resistance include the occurrence of secondary mutations in the TK domain, including the T790M and C797S mutations, and the activation of alternative signaling pathways caused by HER2 amplification or BRAF or PI3K mutations [22,34]. Therefore, durable treatments for NSCLC patients with EGFR activating mutations are currently lacking, underscoring the need for new and more potent therapeutic strategies.

## 4. EGFR Mutations and Implications for Neoantigen Presentation

The most frequent EGFR mutations in different cancer types are illustrated in Figure 2. It is clear that different cancer types have unique mutation patterns. Mutations occurring in GBM are frequently enriched in the extracellular ligand-binding region; for example, the EGFRvIII mutation causing the loss of exons 2 to 7, indicating that sustained downstream signaling caused by ligand-independent EGFR activation may be involved in the pathogenesis of GBM. By contrast, in NSCLC the most common mutations are located within the intracellular TK domain (Figure 2) (e.g., the L858R point mutation or exon 19 E746_A750 deletion).

The amino acid sequence changes caused by genetic mutations during tumorigenesis produces oncoproteins that are present in malignant cells but not in normal cells. These proteins are synthesized, processed and can be presented as peptide antigens by major histocompatibility complex (MHC) molecules on the cell surface of tumor cells as well as dendritic cells (DC) that have processed proteins derived from tumor cells. These peptide/MHC proteins are subsequently recognized by the T cell receptor (TCR) of T lymphocytes, potentially inducing the activation, expansion, and antitumor function of tumor antigen-specific T cells. Activated effector CD8+ T cells that recognize tumor antigens can attack and eradicate tumor cells through perforin and granzyme B-mediated cytolysis, Fas/FasL pathway activation, and/or cytokine release (such as IFN-γ and TNF-α) (Figure 3). These aberrant peptide antigens derived from gene mutations in cancer cells are called neoantigens (NeoAg) or tumor specific antigens (TSA), and are considered immunogenic if they can elicit a T-cell mediated immune response. NeoAgs originating from high-frequency mutations shared by a large number of cancer patients are referred to as public NeoAgs. Another important category of tumor antigens are the so-called tumor-associated antigens (TAA), which are derived from proteins expressed by non-mutated genes that are significantly over-expressed in tumor cells compared with normal cells. Since TAAs can be expressed by normal cells, immunotherapeutic targeting of TAAs carries the risk of on-target off-tumor toxicities. In addition, T cells targeting TAAs often cannot induce sufficiently robust immune responses as a result of central and peripheral tolerance. By contrast, targeting NeoAgs/TSAs that are expressed exclusively by tumor cells should not cause on-target off-tumor toxicities, and the immune response is not confined by tolerance. This makes NeoAgs ideal targets for cancer vaccines and T cell-based immunotherapies. Further evidence of their clinical utility has been demonstrated by immune checkpoint inhibitor (ICI) trials showing NeoAg load to be a predictive marker of clinical response [35]. Due to the growing recognition of the importance of NeoAg in cancer immunotherapy, genomic and bioinformatic approaches to predicting and prioritizing immunogenic neoantigens are flourishing, allowing for the design and direct application of personalized, NeoAg-targeted immunotherapies in cancer patients [36].

The prevalence of shared EGFR mutations in multiple cancer types provides a potentially ideal resource for public neoantigens. Point mutations and insertions create novel peptides that can be presented by MHC molecules. In-frame deletions such as EGFRvIII and exon 19 E746_A750del give rise to newfound proximity of normally distant parts called neo-junctions, which can serve as potential neoantigens. Immunogenicity analysis of the most prevalent EGFR mutations has been performed by multiple research groups to facilitate the development of new immunotherapies.

The EGFR T790M mutation is amongst the most common mutations responsible for acquired resistance to EGFR TKIs, and has been observed in up to half of cases of TKI resistance. By combining bioinformatic peptide binding prediction tools NetMHC 3.2 and BIMAS with in vitro cell-based validation assays, a team from Japan identified two novel HLA-A*02:01-restricted peptide epitopes containing the the EGFR T790M mutation, T790M-5 (MQLMPFGCLL) and T790M-7 (LIMQLMPFGCL). PBMCs from 5 of 6 (83%) and 3 of 6 (50%) healthy donors showed IFN-γ secretion in response to T790M-5 and T790M-7 stimulation, respectively [37]. CD8^+^ T cells reactive to these neoantigens were enriched from these PBMCs and displayed potent and specific cytotoxicity against NSCLC cell lines bearing T790M mutation in an HLA-A*02:01-restricted manner [37]. Another research group published a similar finding [38], showing that the IMQLMPFGC peptide is an EGFR T790M-derived neoantigen bound to HLA-A*02:01. IMQLMPFGC neoantigen-specific CD8^+^ T cells induced from the PBMCs of healthy donors also showed strong recognition and reactivity in vitro to NSCLC cell lines harboring both T790M mutation and HLA-A*02:01 [38]. Both studies suggest that these T790M-derived neoantigens identified might provide a novel immunotherapeutic approach for overcoming EGFR-TKI resistance in NSCLC patients expressing the T790M mutation and HLA-A*02:01.

L858R and E746_A750del, the top two most prevalent mutations in the TK domain of EGFR, can also produce neoantigens. Recently, a team from the University of Colorado used a NetMHCpan 4.0-based in silico prediction followed by in vitro biochemical validation strategy to identify the most frequent mutant EGFR-derived neoantigens bound by prevalent human leukocyte antigen (HLA) class I alleles in NSCLC patients [39]. HLA-A*31:01, A*33:01, B*08:01, and B*27:05 were identified to strongly bind L858R-derived neoantigens but not the corresponding wild-type peptides, whereas HLA-A*11:01 and A*03:01 bound to peptides encompassing the E746_A750del neoantigens with high affinity. These results were supported by the work of a Chinese team that investigated a large cohort of 1862 Chinese NSCLC patients for neoepitopes [40]. Among this cohort, EGFR mutations occurred at a very high frequency, with L858R and E746_A750del being the most dominant, with frequencies of 23% and 13% respectively. Interestingly, the percentage of patients bearing these EGFR mutations along with the HLA alleles predicted to bind to these neoantigens is significant. For example, data mining of the TCGA showed that one-third of patients with early-stage lung adenocarcinoma bear either EGFR L858R or E746_A750del and at least one HLA allele binding neoantigens encompassing these 2 mutations. Within the cohort of Chinese NSCLC patients, L858R and HLA-A*33:03 coexist in 2.93% of patients while E746_A750del and HLA-A*11:01 cooccurred in 5.6% of the NSCLC patients [40]. The coexistence of these EGFR mutations and HLA alleles presenting them in patients may bring some immune protection as these patients exhibited better disease-free (DFS) and overall survival (OS) than other patients, as shown by the analysis of the TCGA data. [40] Collectively, these results imply that EGFR mutations are viable targets for designing neoantigen-based immunotherapies for NSCLC patients.

EGFRvIII, the dominant mutation occurring in around 30% of GBM patients, is characterized by the deletion of *EGFR* exons 2–7, resulting in a removal of amino acids 6–273 from the extracellular ligand-binding domain and inserting a glycine residue not found within the reading frame of wild type EGFR, thus creating a novel junction between exons 1 and 8 [31]. This neo-junction also results in highly immunogenic neoepitopes. PEPvIII (LEEKKGNYVVTDH) is the most well-known neoantigen derived from this neo-junction [41,42], and has been used to vaccinate patients with EGFRvIII-expressing GBM in multiple clinical trials [31,41,42]. However, PEPvIII was originally regarded as a B cell peptide epitope and thus does not require the presentation by MHC molecules. Since the EGFRvIII mutation occurs in the extracellular domain of EGFR, this neoantigen can be directly recognized by specific B cell receptors (BCR) or antibodies, triggering a strong humoral immune response [43]. EGFRvIII-derived MHC-restricted neoepitopes recognized by CD8^+^ T cells have also been identified by in vitro experiments [44,45], implying that this neoantigen also has the potential to induce cellular immunity. This finding was consistent with reports that cancer patients bearing EGFRvIII can spontaneously develop humoral and cellular immune responses against EGFRvIII [46]. PEPvIII-based vaccines were also shown to induce both humoral and cell immunity [47,48].

## 5. EGFR Targeted Peptide Vaccine Studies in Mouse Models

Neoantigen vaccination is a therapeutic strategy where immunogenic polypeptides encompassing tumor-associated mutations are synthesized and administered to patients, thus activating an immune response against tumor cells and providing long-term protection. The high immunogenicity of commonly shared EGFR mutations implied that vaccinating against these EGFR mutations may be a potential strategy in treating cancers harboring these mutations, or preventing cancer recurrence after EGFR TKI therapy.

Rindopepimut (CDX-110) was among the earliest EGFR-targeting vaccines studied in animal models [42]. It is based on the neoantigen PEPvIII (LEEKKGNYVVTDH) derived from the newly formed junction of EGFRvIII, but its peptide sequence has an additional cystine on the C-terminus (LEEKKGNYVVTDHC)in order to link it the keyhole limpet hemocyanin (KLH) carrier protein to enhance immunogenicity [42]. The administration of CDX-110 has been proven to induce the production of antibodies that specifically bind to EGFRvIII expressed on the surface of glioma cells in multiple animal models including mice, rabbits, goats and monkeys [42]. Some of the antibodies isolated from the serum of the vaccinated mice showed potent antibody-mediated cell toxicity against EGFRvIII-expressing or wild-type EGFR-overexpressing tumors in vitro or in vivo [49,50,51].

The direct evidence that CDX-110 prevents tumors in animal models was reported by Heimberger and colleagues from MD Anderson Cancer Center in 2003 [47]. Their mouse model developed tumors after subcutaneous engraftment of EGFRvIII-expressing cancer cells. Peptide vaccination prior to tumor engraftment prevented ~70% of mice from developing subcutaneous tumors and significantly suppressed tumor growth. The vaccinated mice had a prolongation of median survival by 173% in comparison with non-vaccinated controls. For mice that had already developed visible tumors, CDX-110 also prolonged median survival by 26%. Interestingly, in addition to eliciting specific antibodies, vaccination in this mouse model also induced the activation of natural killer (NK) cells and CD8^+^ T cells [47]. The anti-tumor efficiacy of this EGFRvIII-derived neoantigen peptide vaccine was thus interpreted to be a result of cooperation between humoral and cellular immunity.

The antitumor effect of CDX-110 could be further strengthened by combining it with adoptive infusion of dendritic cells (DCs). In a murine brain tumor model established by intracranial injection of EGFRvIII-positive tumor cells, the administration of a DC vaccine pulsed with CDX-110 prior to tumor challenge greatly prolonged survival time, with a 6-fold increase in 62.5% of the vaccinated animals. Since the introduction of DCs mainly enhanced T cell-mediated immunity elicited by CDX-110, the MHC haplotype of tumor cells in these models influenced the efficacy of these DC vaccines. Vaccination in mice bearing an MHC-I haplotype with low CDX-110 affinity resulted only in a mildly prolonged survival time, but mice harboring MHC-I strongly binding to CDX-110 demonstrated a dramatic increase in the median overall survival [52].

Another study indicated that peptides derived from the non-mutated part of EGFR can also protect mice from tumors driven by human EGFR mutation [53]. To model the efficacy of vaccination against EGFR to prevent primary NSCLC resulting from EGFR mutation, a gene-engineered murine model was established to inducibly express full-length human *EGFR* gene with the L858R mutation in their lungs upon doxycycline administration. Without vaccination, these animals rapidly developed lung adenocarcinomas within 10 weeks of doxycycline administration, with 100% tumor incidence.. The vaccine used in this study was comprised two separate peptides along with Freund’s adjuvant. The two peptides contained residues 306–325 (located in the extracellular domain) and 897–915 (located in the intracellular TK domain) of human EGFR, and showed a homology of 80% and 100% between humans and mice, respectively. Interestingly the L858R mutation was not encompassed within these two peptides. However, administration of these peptides elicited a strong and specific immune response in immunized mice. In addition, a dramatic increase of a CD4^+^ CD44^+^ CD62L^+^ central memory T cell population was also seen in vaccinated mice, implying the establishment of long-term immune protection. Impressively, the vaccinated animals showed a 76.4% reduction in tumor multiplicity 12 weeks after induction of the *EGFR* transgene compared with non-vaccinated controls [53]. This indicated that lung tumor development caused by overexpressed human EGFR bearing an L858R mutation can be effectively inhibited in a preventive setting using a multi-peptide vaccine targeting non-mutated regions of EGFR. This study provided potential clinical implications for the development of EGFR-targeted vaccines in NSCLC patients with EGFR mutations.

## 6. Clinical Studies of EGFR Targeted Peptide Vaccines for the Treatment of Cancer

Vaccines directly targeting neoantigens are now being used in clinical trials in various solid tumors [35]. The efficacy of vaccines targeting EGFR mutations in animal models has largely promoted their application in the clinic. In particular, the CDX-110 vaccine has been assessed in GBM patients in multiple clinical trials (Table 1).

Sampson and colleagues reported the first Phase I clinical trial of DCs pulsed with CDX-110 for the immunization of GBM patients. It was a dose escalation and toxicity study that enrolled 12 patients with newly diagnosed GBM who had undergone surgery to remove the brain tumor followed by conformal external beam radiotherapy [48]. A range of 2.7 × 10^7^ to 1.0 × 10^8^ autologous DCs pulsed with CDX-110 were infused into each patient. Adverse events were limited to grade 2 toxicities. Immunization boosted humoral immunity in patients, leading to increased production of EGFRvIII-specific antibodies by B lymphocytes; furthermore, antigen-specific T cells were detected in the peripheral blood in 10 of 12 immunized patients. Median progression-free survival (PFS) from the time of vaccination and median overall survival (OS) from the time of histologic diagnosis was 6.8 months and 22.8 months respectively [48]. This pioneering study was the first to demonstrate the safety and efficacy of an EGFR mutation-derived vaccine for cancer.

To further assess the safety and efficacy of the CDX-110 peptide vaccine, multiple Phase II trials were initiated. In the Phase II trial ACTIVATE, 18 patients with newly diagnosed EGFRvIII-expressing GBM underwent tumor resection, immunization with peptide vaccine, and standard therapy with radiotherapy and TMZ. A matched cohort of 17 patients who received surgery and standard therapy without vaccination were selected for OS and PFS comparison [54]. The toxicity of the vaccine was deemed to be minimal. CDX-110 vaccination in GBM patients prolonged median PFS and OS to 14.2 and 26.0 months, respectively, compared with 6.3 and 15.0 months in the matched control group. Remarkably, loss of EGFRvIII expression was observed in 82% of recurrent tumors in the vaccinated group, highlighting the efficacy of the CDX-110 vaccine [54]. The outgrowth of EGFRvIII-negative tumors may have been due to the existence of other cancer drivers, possibly reflecting tumor heterogeneity. Since it has been reported that lymphopenia can abolish the immune-suppressive properties of regulatory T cells (Tregs) leading to enhanced anti-cancer immunity [55], another Phase II trial (ACTII) was carried out to test whether TMZ-induced lymphopenia reinforced CDX-110 vaccine efficacy in GBM patients [56]. In this study which enrolled 22 EGFRvIII-expressing GBM patients, CDX-110 was administered along with a standard or higher dose of TMZ, which caused transient grade 2 or stable grade 3 lymphopenia, respectively. The vaccinated patients who received a higher dose of TMZ developed augmented anti-tumor humoral immunity compared with those receiving a standard TMZ dose. CDX-110 was also tested in combination with other drugs. In the phase II trial ReACT, EGFRvIII-positive rGBM patients who received CDX-110 and bevacizumab combination therapy were compared with those who had received bevacizumab alone [57]. Patients who received combination therapy had a significantly higher 6-month PFS rate (28% vs. 16%) and 24-month OS (20% vs. 3%) than those patients in the monotherapy group [57]. Another Phase II trial (ACTIII) was performed to confirm the results attained in previous ACTIVATE/ACT II trials with a larger sample size of 65 EGFRvIII-expressing GBM patients [58]. This multicenter, single-arm clinical trial successfully reproduced the positive results of ACTIVATE/ACT II in terms of safety andthe induction of specific humoral immunity and prolonged PFS and OS [58].

The significantly improved PFS and OS compared with historical controls in these Phase II clinical trials encouraged researchers to proceed to a Phase III trial. A double-blind, randomized, international Phase III trial (ACTIV) in patients with newly diagnosed EGFRvIII-expressing glioblastoma was initiated [59]. In the ACTIV study, 745 patients were enrolled and randomly assigned to CDX-110 (*n* = 371) or placebo (*n* = 374) groups. Both groups received standard radiotherapy and TMZ, but only the CDX-110 group were immunized. Unfortunately, this study was terminated due to the unexpectedly low efficiency of the vaccine after the interim analysis. In the final analysis, there was no significant difference in OS for patients with minimal residual disease MRD (defined as <2 cm^2^): median OS was 20.1 months in the vaccinated group versus 20.0 months in the control group. Surprisingly, serious adverse events (SAE) were observed in 7 to 9% of patients in both groups and 16 deaths were caused by adverse events (nine [4%] in the vaccinated group and seven [3%] in the control group), which had not been observed in the previous Phase II trials [59].

Peptide vaccines targeting commonly shared EGFR NeoAgs have also been tested in EGFR-mutated NSCLC patients in phase I clinical trials. In 2016, we reported an 85-year-old Asian patient with stage IV lung squamous cell carcinoma that had progressed after 9 cycles of chemotherapy [60]. Their tumor harbored EGFR L858R, one of the most frequent EGFR mutations among Asian NSCLC patients, but was not responsive to the EGFR TKI erlotinib. Ninety-three non-synonymous somatic mutations were detected from a needle biopsy of the original lung tumor by whole exome sequencing and 5 of these were confirmed using a 508 tumor-associated gene sequencing panel, including EGFR L858R. After bioinformatic prediction of mutation-containing peptide fragments encoded by the five mutated genes, 11 NeoAg peptides predicted to bind to the patient’s HLA allotypes were selected and administed in a saline-based neoantigen peptide cocktail, of which 4 peptides encompassed the EGFR L858R mutation. Eight total doses of vaccine were administered at weekly intervals. No adverse events except for a temporary rash at the injection site were noted after vaccination. PBMC immune monitoring using peptide-stimulated IFN-γ secretion and HLA/peptide tetramer staining showed that the administration of this vaccine induced potent and specific cytotoxic T lymphocytes (CTLs), which were largely focused on the EGFR L858R-derived peptides, especially the HLA-A*3101-restricted H9R9 (HVKITDFGR) peptide. The patient experienced a remarkable regression of multiple lung tumor nodules after peptide vaccination, but a liver metastasis continued to progress. which forced vaccination to be stopped. Finally, he died of liver metastasis shortly afterward [60]. Because the EGFR L858R-derived peptides made the most significant contribution to the antitumor immune response, these NeoAg targets may have contributed to the rapid and dramatic regression observed in multiple lung tumors. The safety and efficacy profile of this vaccine approach suggested that EGFR L858R-targeted peptide vaccines may be a promising immunotherapeutic approach for NSCLC patients harboring EGFR L858R, which accounts for ∼20% of all Asian lung cancer patients [22,40].

Building on the results of this case report, we initiated a Phase I trial of personalized NeoAg peptide vaccination (PPV) for treating NSCLC patients [61]. Twenty-four stage III/IV NSCLC patients who had previously progressed after multiple conventional therapies were immunized with personalized multi-NeoAg peptide vaccines (PPV) weekly. Among them, 16 patients harbored EGFR mutations in their tumors and had progressed after EGFR TKI therapy. Seven of these 16 patients harbored the EGFR L858R mutation, 7 had exon 19 deletions (2 of them also accompanied by the T790M mutation), and another 2 harbored the comparatively rare H773L mutation. Nine of the 16 EGFR-mutated patients continued TKI therapy concurrent with PPV and the other 7 received PPV alone [61].

As expected, the vaccination approach showed a very good safety profile. Other than temporary rash, fatigue and/or fever occurring in three patients, no treatment-related adverse events were observed. In the 24 vaccinated patients, median PFS and OS were 6.0 and 8.9 months, respectively. Within 3 to 4 months of their first dose of PPV, 6 of the 24 patients (25%) experienced a partial response (PR) and 1 (4%) attained a complete response (CR), Notably, all seven objective clinical responders were from the EGFR-mutated cohort, including 4 patients that received concurrent TKI + PPV and 3 patients (including the CR) having received PPV alone. Amongst the 7 responding patients, 5 demonstrated specific CD8^+^ T cell responses against EGFR NeoAg peptides in their peripheral blood during the course of PPV, particularly L858R and T790M.. In addition, we detected enhanced expansion and tumor infiltration of L858R-specific CD8^+^ T cells in the PBMC of one responding patient after the administration of a vaccine that contained five NeoAg peptides that all encompassed the EGFR L858R mutation [61]. To our knowledge, this study is the first report of a NeoAg peptide vaccine inducing objective clinical responses in multiple cancer patients. As with the case report patient, it appeared that the EGFR NeoAgs made the strongest contribution to the clinical responses observed; firstly, all 7 of the clinical responders were patients bearing EGFR mutations; secondly, the most potent T cell responses in peripheral blood were against EGFR NeoAg peptides. This study underscores the strong potential of NeoAg peptide vaccines targeting prevalent EGFR mutations for treating NSCLC patients. However, it will be critical to confirm these promising results in the context of larger, randomized clinical trials in the future.

## 7. Studies of EGFR-Targeted Peptide Vaccines in Combination with Other Immunotherapies

Combination therapies involving EGFR-targeted peptide vaccines along with other immunotherapies have not yet been reported in murine models or human clinical trials. However, it is hoped that improved clinical outcomes will be achieved when EGFR-targeted peptide vaccines are used in combination with immune checkpoint inhibitor (ICI) therapy, which has already been observed with other NeoAg peptide vaccine approaches.

Ott and colleagues reported the results of a Phase Ib trial of personalized NeoAg peptide vaccine (NEO-PV-01) in combination with the PD-1 antibody nivolumab in 82 patients with advanced melanoma, NSCLC or bladder cancer [62]. Nivolumab treatment was initiated at week 0, NEO-PV-01 was administered between weeks 12 and 24, and nivolumab treatment was continued for up to 2 years. The combination therapy was safe, and neoantigen-specific T cell responses and epitope spreading to non-vaccine epitopes were observed in vaccinated patients. The median PFS was 23.5 months, 8.5 months, and 5.8 months in the melanoma, NSCLC, and bladder cancer cohorts, respectively. The median OS was not reached in the melanoma and NSCLC cohorts, and was 20.7 months for the bladder cancer cohort [62]. This study indicated that personalized NeoAg vaccination in combination with ICI therapy is feasible and safe in multiple solid tumor types. The safety and efficacy of NeoAg peptide vaccination in solution or DC-pulsed in combination with PD-1 antibody has also been tested in hepatocellular carcinoma (HCC) and metastatic gastric cancer, with overall promising results [63,64].

Patients with EGFR-mutated NSCLC have generally shown poor responses to ICI therapy, such as antibodies targeting PD-1/PD-L1 [65,66,67,68]. The mechanisms underlying the poor response may include the lower PD-L1 expression, lower tumor mutational burden (TMB), and the immunosuppressive tumor microenvironment (TME) resulting from constitutive activation of the EGFR signaling pathway in EGFR-mutated NSCLC [65,68]. EGFR-targeted NeoAg vaccination works mainly through the activation of tumor antigen-specific CD4^+^ T cells and CD8^+^ T cells. Subsequently, these T cells can specifically recognize and lyse tumor cells harboring these EGFR mutations, achieving a potent and specific anti-tumor response. It is hoped that the expansion and anti-tumor function of these EGFR mutation-derived neoantigen-specific T cells can be augmented and reinvigorated by ICI therapy [69,70], therefore enhancing treatment efficacy. This hypothesis will need to be tested in animal models and clinical trials in the future.

**Table 1 vaccines-11-01460-t001:** Summary of the neoantigens derived from cancer-associated EGFR mutations and associated clinical studies.

NeoAg No.	Epitope	EGFR Mutation	HLA-Restriction	Preclinical Study	Clinical Study	Clinical Trial No.	Trial Status	Reference
1	MQLMPFGCLL	T790M	HLA-A*02:01	NeoAg-specific IFN-γ secretion and cytotoxicity in vitro	N/A	N/A	N/A	[37]
2	LIMQLMPFGCL	T790M	HLA-A*02:01					
3	IMQLMPFGC	T790M	HLA-A*02:01	NeoAg-specific IFN-γ secretion and cytotoxicity in vitro	N/A	N/A	N/A	[38]
4	MQLMPFGSLL	T790M/C797S	HLA-A*02:01	NeoAg-specific IFN-γ secretion and cytotoxicity in vitro; peptide vaccine induced NeoAg-specific -specific CTL responses in mice model.	N/A	N/A	N/A	[71]
5	FGRAKLLGA	L858R	HLA-B*08:01	In silico predictions and in vitro validation by binding assays	NSCLC patients bearing these mutations and corresponding protective HLAs were associated with better prognosis.	N/A	N/A	[39]
6	GRAKLLGAEEK	L858R	HLA-B*27:05					
7	KVKIPVAIKT	E746_A750del	HLA-A*03:01					
8	KVKIPVAIKTS	E746_A750del	HLA-A*03:01					
9	KIPVAIKTSPK	E746_A750del	HLA-A*03:01 HLA-A*11:01					
10	HVKITDFGR	L858R	HLA-A*31:01	In silico predictions, NeoAg-specific IFN-γ secretion and pMHC tetramer staining	The EGFR L858R-derived NeoAg peptides cocktail vaccine was administered in patients with stage III/IV NSCLC and led to remarkable tumor regression and robust immune response.	ChiCTR-INR-16009867	Recruiting	[39,60,61]
11	KITDFGRAK	L858R	HLA-A*11:01					
12	HVKITDFGRAK	L858R	HLA-A*31:01					
13	RAKLLGAEEK	L858R	HLA-A*31:01					
14	LTSTVQLIM	T790M	HLA-C*15:02					
15	LEEKKGNYVVTDH	EGFRvIII	N/A	Peptide vaccine induced humoral and cell immunity in multiple animal models and significantly inhibited tumor growth in mice with brain or lung cancer.	CDX-110 vaccination led to robust immune responses, greatly improved PFS and OS with low-grade toxicity in GBM patients. However, in a Phase III trial (ACT IV), the vaccine did not show sufficient efficacy.	ACTIVATE: NCT00643097 ACT II: N/A Re-ACT: NCT01498328 ACT III: NCT00458601 ACT IV: NCT01480479	All completed	[47,48,49,50,51,52,53,54,55,56,57,58,59]
16	LEEKKGNYV	EGFRvIII	HLA-A*02:01	NeoAg-specific IFN-γ secretion and cytotoxicity in vitro	N/A	N/A	N/A	[45]

## 8. Conclusions and Future Directions

As an RTK, EGFR governs multiple critical biological functions including embryogenesis, development, and tissue regeneration. Commonly shared activating mutations of the *EGFR* gene occur frequently in multiple solid tumor types, potentially providing an ideal source of public NeoAgs to target with immunotherapy. Importantly, the immunogenicity of peptides derived from some of these shared mutations has now been validated in vitro and in vivo in animal models and human clinical trials. Furthermore, EGFR NeoAgs appear to act dominantly in the induction of T cell-mediated immunity in patients and were strongly associated with objective clinical responses, supporting the promising future of EGFR NeoAg-targeted peptide vaccines in treating EGFR-mutated NSCLC patients.

In the future, the efficacy of EGFR-targeting peptide vaccines may be further improved by screening for more immunogenic peptides, introducing more effective adjuvants, or combining with other immunotherapy such as ICIs. The identification of shared NeoAgs also opens the door to other immunotherapeutic approaches such as T cell receptor (TCR)-based therapies. In addition, because the immunosuppressive TME can jeopardize the anti-tumor function of neoantigen-specific T cells on which the efficacy of EGFR-targeting peptide vaccines relies, TME remodeling reagents such as TKIs, metabolism modulators and cytokines that can relieve the immunosuppression within the TME may further enhance the efficacy of these immunotherapies.

## Figures and Tables

**Figure 1 vaccines-11-01460-f001:**
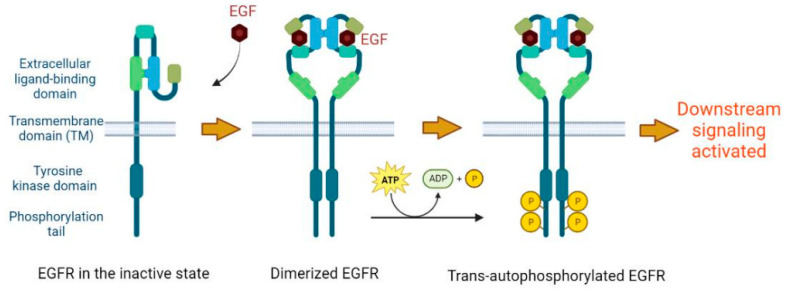
The domain architecture of EGFR and schematic representation of EGFR activation induced by ligand binding. EGFR consists of an extracellular ligand-binding domain, a transmembrane domain (TM), an intracellular tyrosine kinase (TK) domain, and a regulatory tail domain. EGF binding to the EGF receptor unmasks a dimerization motif within the ligand-binding domain and causes structural rearrangements that are conveyed to the cytoplasmic domain, thus allowing the asymmetric dimerization of the two juxtaposed TK domains. The two TK domains allosterically activate each other by trans-phosphorylating the critical tyrosine residues located in phosphorylation tails, thereby triggering downstream signaling.

**Figure 2 vaccines-11-01460-f002:**
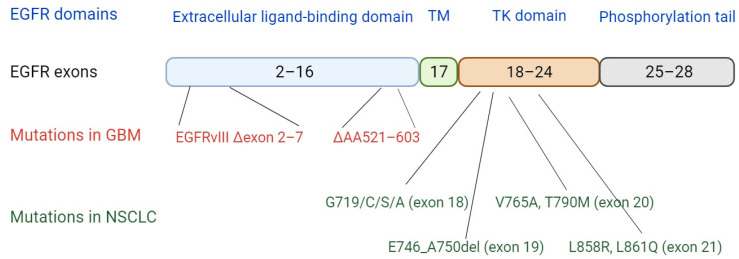
EGFR mutations and their occurrence in different tumor types. The corresponding locations of the mutations occurring within different exons of the EGFR gene are indicated.

**Figure 3 vaccines-11-01460-f003:**
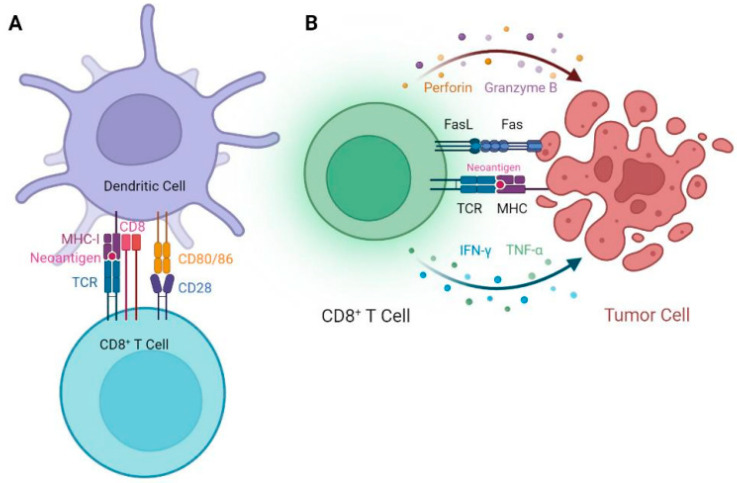
Mechanism of neoantigen-induced CD8+ T cell-mediated antitumor immunity. (**A**) Neoantigens are processed and presented by MHC-I molecules on the cell surface of dendritic cells (DC) and subsequently recognized by the T cell receptor (TCR) of T cells along with the co-receptor CD8. The TCR-MHC-neoantigen interaction, together with the co-stimulation mediated by the interaction between CD28 and CD80/86, induces the activation, expansion and differentiation of antigen-specific effector CD8+ T cells. (**B**) Activated effector CD8+ T cells recognize tumor cells presenting these neoantigens by TCR and attack the tumor cells by perforin and granzyme B-mediated cytolysis, Fas/FasL pathway and cytokines (e.g., IFN-γ and TNF-α) release, which promotes the eradication of tumor cells.

## Data Availability

Not applicable.

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
