# Peer review of "Epidermal Growth Factor Receptor-Targeted Neoantigen Peptide Vaccination for the Treatment of Non-Small Cell Lung Cancer and Glioblastoma"

_vaccines, 2023, doi:10.3390/vaccines11091460_

Round 1
Reviewer 1 Report
The authors discussed the application and the scientific background of the Epidermal Growth Factor Receptor-targeted neoantigen peptide vaccination. The paper is comprehensive, it is detailed enough, but no too detailed and it is easy to understand. The paper gives enough information on why it is an important research field and what are the considerations, when designing a neoantigen peptide vaccine for EGFR.
There are satisfying amount of references included, but more publications from the recent years would increase the impact of the publication.
There are a few typos in the manuscript, but the language is understandable and no intensive editing needed.
With all the above-mentioned notes, I would recommend this manuscript for publication in Vaccines.
There are a few typos in the manuscript, but the language is understandable and no intensive editing needed.
Author Response
August 25, 2023
Dear reviewer,
Thank you for providing valuable comments for our manuscript entitled“Epidermal growth factor receptor-targeted neoantigen peptide vaccination for the treatment of non-small cell lung cancer”(Manuscript submission ID: vaccines-2557263). We have addressed the comments point by point below:
Reviewer #1
1.The authors discussed the application and the scientific background of the Epidermal Growth Factor Receptor-targeted neoantigen peptide vaccination. The paper is comprehensive, it is detailed enough, but no too detailed and it is easy to understand. The paper gives enough information on why it is an important research field and what are the considerations, when designing a neoantigen peptide vaccine for EGFR. There are satisfying amount of references included, but more publications from the recent years would increase the impact of the publication.
Response: To address the reviewer’s comments, we have now replaced References No. 6, 9, 17 and 21 with similar articles published in 2019~2023. Besides, we also added two more recent studies of References 70 and 71 published during 2020~2021 to this manuscript.
2.There are a few typos in the manuscript, but the language is understandable and no intensive editing needed.With all the above-mentioned notes, I would recommend this manuscript for publication in Vaccines.
Response: Thank you for the reviewer’s comments, we have now corrected a few typos on Lines 197, 247, 266, 281, 300, 304, 310, 350, 356, 416, 426, 439, 465, 466 and 478 and highlighted in red in the revised manuscript.
Reviewer 2 Report
Thanks to authors, in my opinion some points need to be re-considered;
1- The figures are not available, at least to me as the reviewer.
2- Have the authors, mainly the corresponding author, have published any original article in this area?
3- I would suggest a table to be added in this manuscript summarizing the ongoing clinical trials along with a brief description of the used NeoAg.
4- Too much discussion is present about glioblastoma. This material may not be necessary since the topic is underscoring the lung cancer.
Author Response
August 25, 2023
Dear reviewer,
Thank you for providing valuable comments for our manuscript entitled“Epidermal growth factor receptor-targeted neoantigen peptide vaccination for the treatment of non-small cell lung cancer”(Manuscript submission ID: vaccines-2557263). We have addressed the comments point by point below:
Reviewer #2
Thanks to authors, in my opinion some points need to be re-considered;
1.The figures are not available, at least to me as the reviewer.
Response: We have now upload the figures again, please let us know if the reviewer can see them.
- Have the authors, mainly the corresponding author, have published any original article in this area?
Response: The corresponding author have published several EGFR associated neoantigen or EGFR related articles as follows:
- Fenge Li, Ligang Deng, Amjad H. Talukder, Kyle R. Jackson, Arjun Katailiha, Sherille D. Bradley, Heather Sonnemann, Qingwei Zou, Caixia Chen, Chong Huo,Yulun Chiu, Matthew Stair, Weihong Feng, Aleksander Bagaev, Nikita Kotlov, Viktor Svekolkin, Ravshan Ataullakhanov, Natalia Miheecheva, Felix Frenkel, Yaling Wang, Minying Zhang, David Hawke, Jason Roszik, Ling Han, Shuo Zhou, Yan Zhang, Zhenglu Wang, Patrick Hwu, Xueming Du, and Gregory Lizée. Neoantigen vaccination induces clinical and immunologic responses in non-small cell lung cancer patients harboring EGFR mutations. J Immunother Cancer 2021;9:e002531
- Li F, Chen C, Ju T, Gao J, Yan J, Wang P, Xu Q, Hwu P, Du X, Lizée G. Rapid tumor regression in an Asian lung cancer patient following personalized neo-epitope peptide vaccination. Oncoimmunology. 2016 Oct 7;5(12):e1238539.
- Li F, Du X, Zhang H, Ju T, Chen C, Qu Q, Zhang X, Qi L, Lizée G. Next Generation Sequencing of Chinese Stage IV Lung Cancer Patients Reveals an Association between EGFR Mutation Status and Survival Outcome. Clin Genet. 2017 Mar;91(3):488-493.
- 3.I would suggest a table to be added in this manuscript summarizing the ongoing clinical trials along with a brief description of the used NeoAg.
Response: To address the reviewer’s comments, a table that provided a concise list of the neoantigens resulting from cancer-associated EGFR mutations and their related pre-clinical and clinical studies has been added to the end of the manuscript, named as Table 1.
4.Too much discussion is present about glioblastoma. This material may not be necessary since the topic is underscoring the lung cancer.
Response: Thanks for the reviewer’s comments. Although our manuscript focused on the EGFR mutation-derived neoantigen vaccines in lung cancer, the most advanced clinical application of these vaccines was also treating patients with GBM. Multiple preclinical studies and clinical trials of CDX-110 against GBM has also provided a lot of invaluable experiences and lessons, which will inspire the future development of EGFR mutation-targeted neoantigen vaccines for patients with non-small cell lung cancer. Therefore, we believed that these information is crucial for the readers to know and it is necessary to discuss the CDX-110 vaccine and glioblastoma in detail. We have also added the words “and glioblastoma” to the end of the manuscript title. Please let us know if the reviewer has any questions for this part.
Reviewer 3 Report
In this manuscript, the authors review the molecular functions of the epidermal growth factor receptor (EGFR), as well as the mutations observed in EGFR in non-small cell lung cancer and glioblastoma. Furthermore, the authors delve into the studies investigating immune responses elicited by neoantigens arising from EGFR mutations. The manuscript also describes both preclinical and clinical studies involving the utilization of neoantigen peptides for cancer vaccination. Overall, this manuscript should be beneficial to readers who are interested in studying neoantigen vaccination for cancer treatment. Comments and suggestions are provided below.
1. Figure 1 and Figure 2 are absent.
2. It would benefit the readers if the authors incorporated a table that provides a concise overview of the neoantigens resulting from cancer-associated mutations, as well as a table summarizing the details of both clinical and preclinical studies.
3. A typo is present on line 338, "ththese".
Author Response
August 25, 2023
Dear reviewer,
Thank you for providing valuable comments for our manuscript entitled“Epidermal growth factor receptor-targeted neoantigen peptide vaccination for the treatment of non-small cell lung cancer”(Manuscript submission ID: vaccines-2557263). We have addressed the comments point by point below:
Reviewer #3
In this manuscript, the authors review the molecular functions of the epidermal growth factor receptor (EGFR), as well as the mutations observed in EGFR in non-small cell lung cancer and glioblastoma. Furthermore, the authors delve into the studies investigating immune responses elicited by neoantigens arising from EGFR mutations. The manuscript also describes both preclinical and clinical studies involving the utilization of neoantigen peptides for cancer vaccination. Overall, this manuscript should be beneficial to readers who are interested in studying neoantigen vaccination for cancer treatment. Comments and suggestions are provided below.
- Figure 1 and Figure 2 are absent.
Response: We have now upload the figures again, please let us know if the reviewer can see them.
- It would benefit the readers if the authors incorporated a table that provides a concise overview of the neoantigens resulting from cancer-associated mutations, as well as a table summarizing the details of both clinical and preclinical studies.
Response: We thank the reviewer’s comments, to address the comment, a table that provided a concise list of the neoantigens resulting from cancer-associated EGFR mutations and their related pre-clinical and clinical studies have added to the end of the manuscript, named as Table 1.
- A typo is present on line 338, "ththese".
Response: The typo presents on line 338 was corrected.
Round 2
Reviewer 3 Report
This reviewer has no further questions.